# Effects of Acute Exposure and Acclimatization to High-Altitude on Oxygen Saturation and Related Cardiorespiratory Fitness in Health and Disease

**DOI:** 10.3390/jcm11226699

**Published:** 2022-11-12

**Authors:** Michael Furian, Markus Tannheimer, Martin Burtscher

**Affiliations:** 1Pulmonary Division, University Hospital Zurich, 8092 Zurich, Switzerland; 2Research Department, Swiss University of Traditional Chinese Medicine, 5330 Bad Zurzach, Switzerland; 3Department of Sport and Rehabilitation Medicine, University of Ulm, 89075 Ulm, Germany; 4Department of Sport Science, University of Innsbruck, 6020 Innsbruck, Austria

**Keywords:** hypoxia, pulse-oximetry, disease, exercise, prevention, therapy

## Abstract

Maximal values of aerobic power (VO_2_max) and peripheral oxygen saturation (SpO_2_max) decline in parallel with gain in altitude. Whereas this relationship has been well investigated when acutely exposed to high altitude, potential benefits of acclimatization on SpO_2_ and related VO_2_max in healthy and diseased individuals have been much less considered. Therefore, this narrative review was primarily aimed to identify relevant literature reporting altitude-dependent changes in determinants, in particular SpO_2_, of VO_2_max and effects of acclimatization in athletes, healthy non-athletes, and patients suffering from cardiovascular, respiratory and/or metabolic diseases. Moreover, focus was set on potential differences with regard to baseline exercise performance, age and sex. Main findings of this review emphasize the close association between individual SpO_2_ and VO_2_max, and demonstrate similar altitude effects (acute and during acclimatization) in healthy people and those suffering from cardiovascular and metabolic diseases. However, in patients with ventilatory constrains, i.e., chronic obstructive pulmonary disease, steep decline in SpO_2_ and V̇O_2_max and reduced potential to acclimatize stress the already low exercise performance. Finally, implications for prevention and therapy are briefly discussed.

## 1. Introduction

The progressively decreasing oxygen supply with increasing altitude is paralleled by an almost linear drop in maximal aerobic power (VO_2_max), the best single measure of the individual cardiorespiratory fitness (CRF). Elevated pulmonary ventilation partly compensates for the reduced ambient oxygen pressure (pO_2_) at high altitude [1]. As the limits of the human pulmonary system may be exceeded even at sea level during intense exercise in highly trained athletes [2], this becomes true for healthy non-athletes [3] and even more for patients suffering from pulmonary diseases exposed to high altitude [4]. Clearly, the compensatory effect of hyperventilation is limited at high altitude, and arterial oxygen saturation (SaO_2_) drops due to the reduced driving pressure for oxygen from the air to the blood and the more rapid transit time of blood across the pulmonary capillary [3]. Consequently, O_2_ delivery to exercising muscles and associated VO_2_max decrease at high altitude, especially when ventilatory requirements are high (e.g., endurance athletes) [5,6,7,8] or in patients with ventilatory limitations (e.g., those suffering from chronic obstructive pulmonary disease [COPD]) [4,9,10]. A close association between pulmonary ventilation, the drop in SaO_2_ or peripheral oxygen saturation (SpO_2_, e.g., measured by finger pulse-oximetry) and VO_2_max is evident and has been repeatedly demonstrated [5,11,12,13,14]. This association is of clinical relevance because the drop in SpO_2_ is not only related to aerobic performance but also to the development of adverse events at high altitude, e.g., acute and chronic mountain sickness [15,16,17,18], acute cardiorespiratory events [19,20,21]. Of course, SpO_2_ change at altitude is not the only modulator of the individual VO_2_max variation but an important and easy to measure one.

The altitude related drop of VO_2_max in subjects with low baseline CRF, particularly in those with pre-existing acute or chronic diseases, can turn even every day activities like walking or stair climbing into high-intensity (close to the individual VO_2_max) physical activities associated with an increased risk for cardiovascular adverse events [16,19,21]. Thus, even small improvements of SpO_2_ (and related VO_2_) during acclimatization to high altitude may increase VO_2_max and reduce the risk of cardiovascular adverse events [10,19,21]. As SpO_2_ can be easily determined at any place, it provides valuable information on changes of the individual VO_2_max (and potentially associated risks) during the high-altitude sojourn [16,17,19].

In contrast to the effects of acute exposures to high altitude or hypoxia on SpO_2_ and VO_2_max, time-, age-, and sex-dependent effects of acclimatization in different individuals (athletes, healthy and diseased persons) have been less considered. Therefore, this review was aimed at evaluating existing evidence on the magnitude of potential benefits of acclimatization on SpO_2_ and related VO_2_max in those individuals, and to provide recommendations on appropriate preparation/acclimatization strategies.

## 2. Materials and Methods

This narrative review is reported following the IMRAD (introduction, methods, results, discussion) format [22] and was not registered at PROSPERO. PubMed, Web of Science and Google Scholar search engines were used to identify appropriate peer-reviewed studies. Search terms such as “oxygen saturation”, “VO_2_max”, “exercise capacity”, “hypoxia”, “altitude”, “acclimatization”, “cardiovascular disease”, “coronary artery disease”, “chronic obstructive pulmonary disease”, “obesity”, “type 2 diabetes mellitus” were used to filter studies related to the topic. According to the objectives, pivotal studies have been hand-picked by the authors.

We here present the results of 3 selected studies, likely reporting representative physiological responses on main determinants of VO_2_max (i.e., maximal values of cardiac output, Qmax; SpO_2_max and related arterial oxygen content, CaO_2_max) in healthy subjects. Further studies have been selected to provide more specific information on those effects depending on the time of acclimatization, initial CRF, age and sex, but also on acute and subacute (acclimatization) responses in patients suffering from chronic cardiovascular and respiratory diseases, i.e., coronary artery disease (CAD), obesity and type 2 diabetes mellitus (T2DM) and COPD. Following, the terms VO_2_max and VO_2_peak, and SpO_2_max and SpO_2_ are used interchangeably as reported in the respective studies.

## 3. Results

### 3.1. Effects of Acute and Chronic Exposure to High Altitude on SpO_2_max and Related VO_2_max Subsection

#### 3.1.1. Healthy, Well-Trained Individuals

Findings on physiological responses to maximal exercise of healthy and well-trained individuals, focusing on main determinants of V̇O_2_max, performed at acute and chronic exposure to altitudes of 4300 m and 5260 m, are presented in Table 1.

The amount of VO_2_max decline (change from sea level; −30%) is comparable with the SpO_2_max decline (−29 to −26%) in acute altitude of 4300 m, but the VO_2_max decline (−46%) is considerably greater than the SpO_2_max decline (−29%) at 5260 m, which is accompanied by a more pronounced reduction of Qmax (19.5 vs. 19 L/min at 4300 m and 23 vs. 19 L/min at 5260 m). With an acclimatization duration over 15 days up to 9–10 weeks V̇O_2_max improvements between 5 and 13% have been reported (Table 1). These improvements were associated with elevation of both the SpO_2_max values and hemoglobin concentration (Hb). While Qmax, SpO_2_max and CaO_2_max did not fully recover (to baseline) during rather short-term acclimatization (15 to 18 days), CaO_2_max (but not Qmax) did during the 9–10-week acclimatization period. The further decrease of maximal heart rate (HRmax) from acute to chronic altitude contributed to the Qmax decrease reported in Table 1.

#### 3.1.2. Patients Suffering from Coronary Artery Disease

It has been demonstrated that VO_2_peak and SpO_2_ values in elderly individuals (with and without CAD) when exposed to moderate altitude (2500 m) declined at a similar rate as known for young and healthy individuals [19,26]. In the study by Levine and colleagues, VO_2_peak values decreased by 12% (24.0 ± 1.2 to 21.1 ± 1.2 mL/min/kg) when acutely exposed to 2500 m but returned to baseline (24.2 ± 4.8 mL/min/kg) after 5 days of acclimatization [19]. Sea level VO_2_peak and the ventilatory response to hypoxia (HVR) explained 92% of the variance in VO_2_peak measured at altitude [19]. Schmid et al. reported a similar altitude-related VO_2_peak decline (28.3 ± 4.4 to 22.9 ± 3.9 mL/min/kg) in CAD patients exposed to 3454 m [27]. Exercise responses at altitude, i.e., ventilation, heart rate, and lactate concentrations were all significantly higher at comparable levels of exercise [27], indicating elevated exercise stress and cardiovascular risks in CAD patients ascending acutely to high altitude [27,28]. Importantly, those studies also demonstrated the ability of normal acclimatization in CAD patients within about 5–10 days [19,29]. De Vries and colleagues found similar changes in exercise performance and related physiological responses in CAD patients and healthy controls, who travelled from sea level to 4200 m for a 10-day acclimatization period [29]. In that study, maximal power (watts) had decreased from sea level to 4200 m by about 30% in CAD patients and controls as well [29].

#### 3.1.3. Obese People and Patients Suffering from Type 2 Diabetes Mellitus

Mountaineering activities are characterized by walking/climbing uphill in a hypoxic environment. Therefore, it is evident that obese people will have harder to work and breath, and will likely become rather hypoxic than their lean and healthy counterparts [29]. Ri-Li et al. showed more rapidly increasing acute mountain sickness (AMS) scores during the simulated high-altitude (3658 m) sojourn in obese compared to non-obese men, and this observation was related to lower SpO_2_ values (median: 75% vs. 83%) in the obese individuals during sleep in hypoxia [30]. Studies on exercise performance and acclimatization in obese people are scarce, but recent data indicate that higher body mass is associated with lower SpO_2_ values in long-term high-altitude residents [31]. Although mild obese people showed even increased chemosensitivity, oxygen desaturation was exaggerated during sleep at high altitude (3568 m) [30,32]. As T2DM is closely related to obesity, the increased fat mass and impaired ventilation may primarily contribute to potential oxygen desaturation and loss in exercise performance at high altitude. Individuals suffering from T2DM may have similar adaptations, e.g., with regard to ventilation and cardiovascular responses, but elevated counter-regulatory hormones may negatively affect glycemic control [33].

#### 3.1.4. Patients Suffering from COPD

In contrast to CAD patients, the SpO_2_ and related VO_2_peak decline in COPD patients is expected to be much more pronounced due to ventilatory restrictions. In patients with COPD, the literature on acclimatization effects on SpO_2_ and exercise performance is scarce. Studies during acute exposure to hypobaric hypoxia at 1650 m showed a change in V̇O_2_peak from 78% at 490 m to 65% predicted at 1650 m. This was associated with reduced SpO_2_peak from 95% to 90%, respectively [34]. In the same study, COPD patients performed submaximal six-minute walk tests after the first and second night at 1650 m and 2590 m, respectively. It was found that six-minute walk performance initially decline by 3% to 5% at 1650 m and 7% to 9% at 2590 m. However, their performance significantly improved even after the first night at both altitudes, despite unremarkable improvements in SpO_2_peak at end-exercise [35]. In contrast, after the second night at 2590 m, during another submaximal exercise test until exhaustion on a semi-supine bicycle exercise test at 60% of maximal performance from 490 m, patients showed a median decline in endurance time of 54% [10]. This reduction in endurance time was associated with SpO_2_max reduction from 92% at 490 m to 81% at 2590 m, which was associated with cerebral hypoxia. Therefore, the magnitude of high altitude-related exercise intolerance seems to depend on the type and exercise intensity in patients with COPD. This has been confirmed by another study conducted at 2048 m in patients with moderate to severe COPD [36]. In this study, 32 patients with COPD performed a maximal exercise ramp test on the day of arrival at 2048 m. The mean VO_2_peak declined by 7% at 2048 m compared to 490 m, which was associated with a SpO_2_peak decline of 8%. In the same study and identical to the study at 2590 m, submaximal exercise performance on a semi-supine bicycle was performed after the first night at 2048 m. Similar to 2590 m, endurance time was strongly reduced by 36% compared to 2048 m; SpO_2_peak declined by 7% [37].

Patients with mild to moderate COPD were less susceptible to altitude-related hypoxia when performing a maximal cardiopulmonary exercise test using a ramp protocol at 3100 m [38]. This has been shown in 50 COPD patients, presenting an altitude-induced reduction in VO_2_peak of 9%, which is similar to patients with more severe COPD at 2590 m.

## 4. Discussion

The presented findings once more emphasize the importance of SpO_2_ on VO_2_max, especially during acute but also during chronic altitude exposure. The close association between the decrease in SpO_2_ and the decrease in V̇O_2_max, explaining more than 70% of the VO_2_max loss at acute moderate and high-altitude (up to about 4500 m), is well established [12,39,40]. Thus, appropriate ventilatory acclimatization is of utmost importance. Even small improvements of SpO_2_ values at rest and at peak exercise, commonly occurring during the first days of acclimatization to high altitude [19,23,24,25,26,41], are associated with VO_2_max improvements [19,23,24,25,26,41] and thus, with reduced risk for major cardiovascular events [42,43,44].

Low SpO_2_ values may result from certain anthropometric characteristics (small lung size), relative hypoventilation and diffusion limitation [7]. Each additional 1% reduction of SpO_2_ (during maximal exercise) below 4% in men and 3% in women, is associated with a 1% loss in VO_2_max even at sea level [45], and may be aggravated during intense exercise at high altitude [5,6,7,8]. Consequently, athletes suffering from exercise-induced hypoxemia (EIH) and patients suffering from ventilatory limitation, are particularly prone to a large decrease of SpO_2_ and associated V̇O_2_max impairment at altitude [8,9,10,46,47]. However, prolonged stay at high altitude favors acclimatization, e.g., including hyperventilation, hemoconcentration and erythropoiesis, contributing to some (but important) recovery of SpO_2_, CaO_2_, VO_2_max and exercise performance [1,48,49,50,51,52].

Beneficial effects of acclimatization occur to a similar extent in healthy individuals and those suffering from cardiovascular diseases [19,26,28]. Those effects remain unclear in obese people and those suffering from COPD, likely related to large differences in individual responses to hypoxia and exercise [31,35].

Regarding appropriate ventilatory acclimatization (hyperventilation), an at least 2-week period of altitude sojourn will be necessary for the prevention of a too marked arterial desaturation during intense aerobic exercise [23,41,53,54,55]. Such acclimatization effects are not only important for healthy and diseased high-altitude tourists and trekkers, but especially for those participating in high-altitude endurance competitions and/or climbing extreme altitudes (like the Mt. Everest) [55,56]. Potential benefits on V̇O_2_max from prolonged sojourns at high altitude are less clear. After about 2 weeks at altitude, erythropoiesis contributes to the increase of hemoglobin concentration, which initially occurs due to loss of plasma volume (hemoconcentration) [57,58,59,60]. The elevation of blood hemoglobin concentration during prolonged acclimatization to high altitude (e.g., 9 weeks at 5260 m) seems not to play an important role on VO_2_max change, as Calbet et al. nicely demonstrated that neither the increase in hemoglobin concentration nor hemodilution affected pulmonary or leg V̇O_2_max, due to reciprocal changes in leg blood flow via local metabolic control of the muscle vasculature [61].

At altitudes above 4500–5000 m, the reduction of Qmax may significantly contribute to the V̇O_2_max decrease. In contrast to moderate altitude, where the reduction of V̇O_2_max was primarily explained by the diminished CaO_2_, the low inspiratory pO_2_, the impaired gas exchange and the reduced Qmax, associated with a reduced leg blood flow, seem equally contribute to the drop in V̇O_2_max in severe hypoxia [62]. The drop of Qmax in more severe hypoxia has been attributed to the reduced cerebral paO_2_ (resulting in HRmax reduction) for instance, by modulation of the output drive from cardiovascular nuclei in the central nervous system [62]. When cerebral hypoxia improves during acclimatization, Qmax would also improve and reinforce the association between SpO_2_ and V̇O_2_max. Again, V̇O_2_max at altitude is best predicted by baseline (sea level) VO_2_max.

### 4.1. Correct Measurement of SpO_2_ (and VO_2_max)

SpO_2_ measured by pulse oximetry is widely used at high altitude to monitor the well-being of climbers or in studies [63]. At sea level, pulse oximetry is an established and very easy-to-use measurement technique; at altitude, measurement is challenging because significant possibilities for error (i.e., hyperventilation, cold and motion artifacts) must be considered [64,65]. The cyclic course of saturation typical at altitude makes pulse oximetry measurement challenging and time-consuming [66] (Figure 1).

Studies should therefore always specify how SpO_2_ values were determined; differences might explain at least a part of the partly contradictory results in studies [15,16,17,67,68,69,70]. At an altitude of 3100 m, the initially observed fluctuations of SpO_2_ diminished after full acclimatization and the determination of SpO_2_ is similar to that at sea level [66]. Recommendations how to measure SpO_2_ correctly at high altitude are given by Tannheimer et al. [66]. The interpretation of SpO_2_ values at high altitude requires a high level of experience.

At altitude paO_2_ and SaO_2_ decrease considerably. In mountaineers on Mount Everest, values for paO_2_ of only 19.1 mmHg and for SaO_2_ of only 34.4% were determined; blood taken by puncture of the femoral artery [71]. This is far below the measurement range pulse oximeters are designed for. Therefore, some authors criticize their use at high altitude because of their potential inaccuracy in the saturation range below 75% [63,65]. Nevertheless, systematic investigations could demonstrate a minimal variance of only 1.8% as low as a SpO_2_-value of 57% [72]. Earlier studies showed a well correspondence in dogs in the saturation range between 22–100%: R² = 0.97 [73], in children in the range between 35–95%: R² = 0.90 [74] and as well in children in the range between 30–80%: R² = 0.94 [75]. In addition, our own study in 14 intensive care patients could demonstrate an excellent correlation between pulse oximetry and bloody determination of SaO_2_ in the saturation range between 45–98%, R^2^: 0.99; *p* < 0.001 [70].

For the measurement of VO_2_max at altitude, the lower air density and environmental pressure that leads to lower alveolar pressure and therefore lower oxygen diffusion in the lung have to be taken into account. A recent study [76] demonstrates during submaximal spiro-ergometry a higher breathing frequency and higher minute ventilation at 4200 m compared to 2800 m, while VO_2_ remained the same. Since physiological measurements in altitude situation are nowadays frequently done in normobaric hypoxia (NH) for reasons of simplicity and cost savings, this study focused on the differences between NH and hypobaric hypoxia (HH), reporting significant differences. When comparing NH to HH at 2800 m and 4200 m, breathing frequency and minute ventilation were higher and VO_2_ was lower in HH [76]. Such differences between NH and HH have to be considered.

In summary, pulse oximetry is easy to use, may provide valuable information on individual responses to high-altitude exposure (acute and during acclimatization) but requires experienced personnel to interpret the findings.

### 4.2. Effects of Baseline Aerobic Performance

At altitude, pO_2_ parallels the decreasing barometric pressure. As a consequence, hyperventilation and elevated cardiac output due to sympathetic activation occur [77], but above 2500 m they are no longer able to completely compensate for the altitude-induced hypoxia [78,79]. SpO_2_ levels decrease with acute altitude/hypoxia exposure and partly recover during acclimatization [77]. In contrast to VO_2_max, the initially reduced submaximal endurance performance improves during 2 to 3 weeks of acclimatization [80,81].

In trained and untrained individuals, there is considerable inter-individual variability of the altitude-induced reduction of aerobic performance [11,78,82]. Although the reasons for these individual responses are not entirely clear, it seems that fitness level may be an important factor, as endurance-trained athletes (VO_2_max > 60 mL/kg/min) have demonstrated a larger decline in VO_2_max with increasing altitude compared with untrained individuals [11,83]. It has been suggested that this is due to the fact that these athletes have developed exercise-induced desaturation already at sea-level [84,85,86] and operate at the steeper part of the oxygen equilibrium curve even at low altitudes [12,83]. The decline of both VO_2_max and submaximal performance in athletes as well as in non-trained or only slightly trained individuals [87] underlines that for demanding physical activities (mountaineering, cycling, skiing, mine workers, etc.) a solid basic (sea level) endurance capacity is necessary to be able to perform such activities at high altitudes.

The importance of CRF (baseline VO_2_max) to perform physical activities, e.g., level walking or uphill walking, is depicted in Figure 2. It demonstrates that people even with a VO_2_max of about 30 mL/min/kg will likely be unable to climb an altitude distance of 200 m/h above 3500 m and those with a VO_2_max below 20 mL/min/kg cannot afford that even at low altitude.

For rescue operations at altitude a rescuer’s pulse work capacity at a heart rate of 170/min (PWC170) of at least 3 watts/kg measured at sea level is necessary [89]. Better trained individuals will still be able to perform at higher absolute exercise intensities for a longer period of time at high altitude. Moreover, when exercising at the same intensity, fitter individuals compared to less trained individuals may experience less fatigue which may prevent adverse effects at altitude [52]. The physiological profile of world-class altitude climbers corresponds to that of endurance-trained individuals [90]. Noteworthy, a recent case report on Kilian Jornet, the elite athlete with the fastest ever known time to the summit of Mount Everest (26.5 h, from Rongbuk monastery, 5100 m) reported a VO_2_max of 92 mL/kg/min (measured on May 08, 2010, CAR Sant Cugat, Spain, using cycle ergometer) with a maximal heart rate of 199 bpm and maximal minute ventilation of 199 L/min [91]. Taken together, baseline VO_2_max predicts endurance performance at high altitude, and due to the close association between VO_2_max and SpO_2_, any improvement of SpO_2_ during acclimatization may be helpful, especially in individuals with low baseline VO_2_max (see Figure 2).

### 4.3. Effects of Age

The aging process is associated with an unavoidable decline in exercise performance, becoming more relevant at high altitude due to the altitude-related decrease in V̇O_2_max [40,88,92], but can be largely modified by the maintenance of individual training habits [93]. V̇O_2_max decreases with increasing age mainly because of the decrease in Qmax and less because of changes in the arterio-venous oxygen difference, since the enzymes of oxidative energy metabolism as well as capillarization do not markedly change with age. There are two reasons for the reduction of Qmax, first, maximal HR decreases by about 0.7 beats per minute per year, and second, the stroke volume decreases to 80–90% of the values of young people. Of interest in this context is that V̇O_2_max is trainable across all age groups. Storen et al. could show that participants from 20 to 70+ years of age revealed similar improvements in V̇O_2_max and that the training response to short-term high-intensity interval training was not affected up to moderate age in individuals with a representative V̇O_2_max for what is typically observed in the population. Their results showed that the magnitude of V̇O_2_max improvement was affected by the initial training status [94]. Beside the general decline in V̇O_2_max, several aging-related changes may negatively affect high-altitude performance in the elderly, e.g., the development of cardiovascular and pulmonary diseases [88], loss in vision and hearing [95] or changes in blood rheology [96]. In summary, altitude related changes in V̇O_2_max (and SpO_2_max) may become increasingly important in the elderly due to the declining V̇O_2_max with aging.

### 4.4. Effects of Sex

In a review with data derived from seven studies, Fulco et al. found no difference between men and women in percent VO_2_max decrement with increasing altitude (580–6000 m); both sexes showed a similar VO_2_max decline with increasing altitude compared to 67 other studies without consideration of sex [87]. However, due to mechanical ventilatory constraints, appropriate hyperventilation seems to be more limited in females than males, likely resulting in a more pronounced decline in SpO_2_ and associated VO_2_peak in women [97,98]. These assumptions have been confirmed by Woorons and et al. [99]. Horiuchi and colleagues also showed that SpO_2_ in females compared to males was more affected by exercising minute ventilation and energy expenditure [97]. Thus, a steeper fall in VO_2_peak with increasing altitude may be expected in women, which may become especially challenging when climbing in extreme altitudes [100]. However, previous studies have not assessed or considered the influence of pre/post menopause, hormonal fluctuations across the menstrual cycle and hormonal contraception; therefore, further studies are needed to conclusively determine the relationship between SpO_2_ and V̇O_2_max at altitude in women. In summary, SpO_2_max might more decline in women sojourning to high altitude, but confirmation of this observation is necessary.

### 4.5. Effects of Pre-Existing Cardiovascular and Metabolic Disease

Appropriate CRF is of particular importance when sojourning to high-altitude regions due to the sloped terrain and the hypoxic environment. Requirements (regarding VO_2_peak) increase about 2.5-fold when walking at a speed of 3 km/h at a gradient of 15% compared to level walking at the same speed [51] (see also Figure 2). In addition, physiological responses to exercise (e.g., ventilation, heart rate, blood pressure) are elevated at high altitude [101], and individual VO_2_peak levels decrease by about 10% every 1000 m of additional increase above 1500 m above sea-level [51,81]. Thus, low CRF constitutes an important risk factor in susceptible individuals, e.g., for severe cardiovascular adverse events during exercise in general [102], and specifically when performing physical activity during the first days at high altitude [42,44]. Hypoxemia (even it is not more severe than in healthy individuals) remains a major risk in CAD patients ascending to high altitude, because the elevated myocardial oxygen demand (due to increases in heart rate and systemic blood pressure) may exceed myocardial oxygen delivery [27,103]. If this is the case, myocardial ischemia may not only force patients to stop exercise but may also provoke severe cardiac adverse events [19,42,103].

Increases of HR, cardiac contractility, and Q, accompanied by elevation of myocardial workload and oxygen demand, are major adjustments in acute altitude conditions [104,105]. Resting and exercising HRs increase in parallel with the reduction of SpO_2_, due to sympathetic activation and vagal withdrawal [59]. Linked to this, systemic blood pressure increases during acute high-altitude exposure, especially important in hypertensive subjects [106]. As oxygen extraction is already high at low altitude, the myocardium almost exclusively relies on coronary vasodilatation and enhancement of coronary blood flow, which may become limited particularly in patients suffering from CAD but will improve to some extent with acclimatization [59,104,107,108].

Acclimatization (e.g., hyperventilation and hemoconcentration) to high altitude improves hypoxia tolerance and physical performance over time, associated with elevated ischemic threshold and diminished arrhythmia frequency in patients with preexisting CAD [19,26]. Beneficial effects of even short-term acclimatization or hypoxia preconditioning may develop, based on the observation that the risk of sudden cardiac death steeply decreases when spending the first night closer to the altitude where physical activities (e.g., mountaineering, skiing) on the following day will be performed [44]. At least in healthy subjects (but also indicated for elderly and those suffering from CAD [19]), the initially (acute exposure to 3200 m) diminished cycling performance (i.e., 50-min time trial) improved by 5.7% after short-term acclimatization (2 days and nights at altitude) [41]. In this study, the accompanied increase of SpO_2_ explained 97% of the variance of the related performance improvement [41]. Furthermore, low SpO_2_ values after a 6-min walk test (6MWT), a single measure to assess the functional status in patients with cardiovascular disease, were predictive for successfully reaching the summit of Aconcagua (6961 m) [109]. These authors demonstrated a 97.2% sensitivity of a post-exercise SpO_2_ < 75% (measured at Plaza de Mulas base camp, 4365 m) in predicting the outcome of successfully reaching the Aconcagua peak [109]. Appropriate pharmacological treatment of systemic hypertension (e.g., with angiotensin receptor blocker-calcium channel blocker combination) was shown to be effective and safe when sojourning to high altitude [107]. However, certain medication (e.g., beta-blockers) may negatively affect SpO_2_ and/or exercise performance [110,111]. Finally, there are several other factors associated with SpO_2_ and performance, and the risks of AMS and/or cardiovascular adverse events at high altitude, e.g., natriuretic peptides (NPs: BNP/NT-proBNP) are related to both high pulmonary artery pressure and AMS as well [112].

Obesity and type 2 diabetes (T2DM) are often linked with low CRF and reduced pulmonary function [113], likely adversely impacting on both aerobic exercise performance at and acclimatization to high altitude. The excess of adipose tissue is associated with a reduction in lung volume and impaired respiratory mechanics [114], fostering relative hypoventilation and oxygen desaturation during exercise at high altitude. Interestingly, chemosensitivity was demonstrated to be even exaggerated (not blunted) in obese individuals, but during the night time SpO_2_ dropped considerably more in obese compared to normal weight individuals [32], probably contributing to the explanation of increased risk for AMS in obese people [115]. The risk of sleep-disordered breathing is elevated in obesity, and excess body mass importantly predicted the severity of arterial oxygen desaturation during apnea and hypopnea events [116]. In addition, during fatiguing exercise, the high-altitude environment may provoke more severe hypoxemia and performance limitation in obese compared to normal-weight individuals [117]. Nevertheless, physiological adjustments to hypoxia exposure seem not to be compromised in non-obese T2DM patients, provided they are physically fit and well-prepared (with appropriate diabetes self-management skills), they will tolerate high-altitude sojourns as well as do their healthy peers [33].

Taken together, while patients suffering from CAD or T2DM show rather normal adaptations to high altitude, severely obese individuals seem to be more negatively affected. Importantly, an appropriate CRF (depending on the level of physical activity and altitude expected) is of utmost importance for all these patients intending to visit high-altitude regions, and pulse oximetry (SpO_2_ and heart rate values) will provide valuable information on acute responses and the progress of acclimatization to high altitude.

### 4.6. Effects of Pre-Existing Respiratory Diseases

As described above, exercise at altitude is physically more demanding than exercise at low altitudes. Respiratory diseases such as COPD, cause exercise limitations even at low altitude with gas exchange impaired and for some individuals, desaturation even during submaximal exercise [118]. Here we primarily focus on COPD because a large number of studies has been published on exercise (in)tolerance of these patients at sea level and high altitude as well. A study with lowlanders suffering from COPD (Gold stage 2 and 3) ascending to 1650 m and 2590 m showed a significantly reduced distance in 6 MWT and SpO_2_ (measured at beginning and end of the 6MWT), both compared to baseline (490 m) [34]. The participants in this study stayed at respective altitude for one night and repeated the 6MWT on the next day, which revealed an improved distance on the second day. The SpO_2_, which decreased significantly with altitude, did not improve over the night spent at altitude. It seems one night of acclimatization improved CRF but did not yet have a significant impact on the oxygen saturation. Another study done in high-altitude natives suffering from COPD and a healthy control group at 2640 m showed a lower exercise capacity of COPD patients and the exercise capacity worsening with more severe COPD [119]. In this study a statistically significant lower arterial oxygen saturation was noted in the COPD patients than in the healthy control. This is in line with the statement that exercise capacity and CRF are closely linked to the arterial oxygen saturation and CRF decreases if SpO_2_ decreases. Based on COPD studies at altitude examining maximal exercise performance, it seems that more severe COPD patients suffer from more altitude-related exercise intolerance compared to less severe COPD. Whether these impairments are related to lower pre-exercise SpO_2_, stronger EIH due to disease-related ventilatory constrains, or other factors remains to be elucidated. However, interestingly, reduction in maximal exercise performance with altitude, seems to correspond to expected reductions in healthy when travelling to high altitude, with the only difference, that COPD patients present a markedly reduced baseline exercise performance compared to healthy. However, even submaximal exercise performance in COPD seems to be strongly affected by high altitude. This has also been seen by more pronounced EIH, which might be a strong predictor for exercise endurance performance in COPD at altitude.

Pulmonary hyperinflation represents the major limitation to exercise performance in COPD [120], which is closely associated with exertional oxygen desaturation in these patients [121]. Thus, intense exercise at high altitude may not only compromise COPD patients due to an exaggerated SpO_2_ and V̇O_2_max decline but even older athletes suffering from some ventilatory constraints [9].

Of note—apart of the reported exercise intolerance in patients with COPD, these patients are prone to altitude-related adverse health effects requiring medical intervention or premature descent to lower altitudes. The incidence of these unfavorable effects has been reported to be up to 76% at altitudes of only 3100 m [122]. When taking into account this knowledge, that the majority of COPD patients are not able to perform any physical exercise due to adverse health effects, then the altitude impact on exercise performance is currently underestimated and largely unknown in COPD at altitude.

Only to mention, in a study from Schneider et al. [123], the effect of spending several hours at 2502 m on the duration of a constant work-rate exercise test in participants with pulmonary hypertension was investigated. Similar to patients with COPD, exercise performance was strongly reduced at high altitude and was associated with lower SpO_2_max (94 ± 2% at 490 m vs. 87 ± 8% at high altitude).

Taken together, decreases of SpO_2_ and exercise performance are more pronounced in COPD patients exposed to acute and chronic high altitude when compared to healthy individuals and those with cardiovascular and/or metabolic diseases.

## 5. Limitations

As outlined in the methods, insufficient and heterogenic literature was identified to quantitatively (meta-analysis) assess the relationship between SpO_2_ and VO_2_max during acute and chronic exposure to high altitude in these populations. Therefore, this narrative review summarizes findings from many highly demanding and unique but small altitude studies, and further studies are warranted to consolidate our knowledge related to hypoxia and exercise. In particular, more insights for the SpO_2_—VO_2_max relationship would be desired during chronic altitude exposure in patients with pre-existing diseases; the impact of pre / post menopause, menstrual cycle-related hormone fluctuations and contraception in women; and the impact of exercise modalities on the relationship between changes in SpO_2_ and its impact on VO_2_max and endurance time.

## 6. Preventive and Therapeutic Implications

All measures resulting in improved SpO_2_ and exercise performance when exposed to high altitude will contribute to the increase in well-being and the prevention of adverse events during the high-altitude sojourn. Such measures include appropriate physical preparation (fitness training), pre-acclimatization (altitude/NH exposures) and/or (if necessary) pharmacological prophylaxis, e.g., use of acetazolamide), and individually adapted behavior (e.g., rest or low-intense physical activity, continuation with acetazolamide) at high altitude. In addition, the trained use of pulse oximetry may be helpful to track individual acclimatization. Pre-travel exposures to moderate terrestrial altitude and or exposures to progressive intermittent NH are of particular relevance not only for pre-acclimatization but also to experience individual responses to high altitude at rest and during exercise (for more detailed information see [124,125,126,127]). The attending physician is especially important for risk stratification and appropriate counselling, although with regard to treatment options in the case of adverse events at high altitude [101].

As NH rooms and tents have become increasingly available in recent years, NH challenge testing (HCT) may become helpful especially in COPD patients for the prediction of oxygen desaturation and exercise performance at the target altitude. The HCT, initially developed to asses tolerance of air travel where patients are exposed at rest for 20 min to an inspiratory oxygen fraction of 15% (simulating an altitude of 2438 m) [127], can be individual adapted according to the expected conditions (e.g., exercise modality and temperature) and the altitude exposed. While in COPD, acetazolamide (e.g., 375 mg/day) will help to prevent adverse health effects, nocturnal hypoxemia, sleep apnea and improve subjective sleep quality at altitude [125], dexamethasone treatment (4 mg bd) may also be considered but has been associated with hyperglycemia in these patients [128]. In addition, several practical issues need to be considered in COPD patients to make a shared decision on the planned travel to high altitude, e.g., availability of supplemental oxygen and medical on-site management. In some cases, abandon of the high-altitude journey has to be considered.

## Figures and Tables

**Figure 1 jcm-11-06699-f001:**
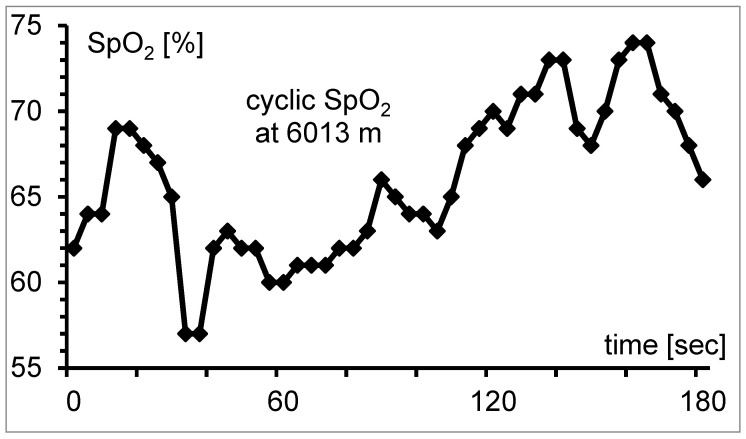
Cyclic course of SpO_2_ at 6013 m because of periodic breathing. It is difficult to determine the mean value of the measurement interval from the instrument display. The gap between minimum and maximum is 17%-points; mean SpO_2_ is 66.0%.

**Figure 2 jcm-11-06699-f002:**
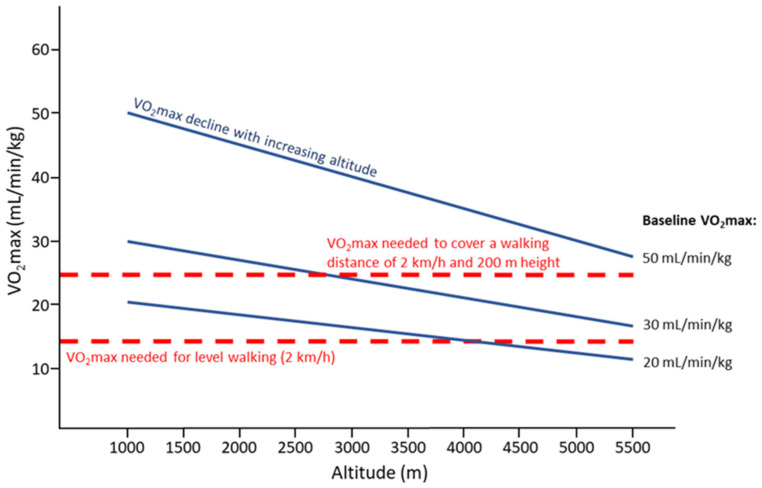
Decline of VO_2_max with increasing altitude (solid lines) depending on the baseline VO_2_max (50, 30, and 20 mL/min/kg), and VO_2_max needed for level walking at a pace of 2 km/h or covering 2 km/h on a slope with a 200 m difference in altitude (dashed lines); based on calculations provided in [88]. It is assumed that people will walk at 60% of their individual VO_2_max.

**Table 1 jcm-11-06699-t001:** Changes of VO_2_max and its determinants in well-trained subjects exposed to acute and chronic high altitude.

Reference	N, Sex	Mean Age, Years	Altitude, m	Exposure Duration	Sea Level (SL)		Acute Altitude (AA)		Chronic Altitude (CA)		
					VO_2_max mL/min/kg	SpO_2_max %	VO_2_max mL/min/kg (Change from SL)	SpO_2_max % (Change from SL)	VO_2_max mL/min/kg (Change from SL; from AA)	SpO_2_max % (Change from SL; from AA)	Change in Ventilation [Hb or Hct] from AA
					HRmax, bpm Q, L/min	CaO_2_ mL/dL	HRmax bpm Q, L/min	CaO_2_ mL/dL	HRmax bpm Q, L/min	CaO_2_ mL/dL	CaO_2_, % SL
Horstman et al., 1980 [23]	9 males	21	4300 m	15 days	50.6	94.5	35.3 (−30%)	69.7 (−26%)	38.8 (−23%); (+10%)	74.2 (−21.5%) (+6.5%)	+10% [+13%]
					174 19.5	22.1	167 19	16.8	161 17	19.9	90%
Beidleman et al., 1997 [24]	6 males	31	4300 m	18 days	57.0	97.0	40.0 (−30%)	69.0 (−29%)	42.0 (−26%) (+5%)	73 (−25%) (+6%)	+11% [+11%}
					190 ---	19.8	177 ---	14.2	170 ---	16.7	84%
Calbet et al., 2003 [25]	4 males 3 females	24	5260 m	9–10 weeks	56.3	95.9	30.0 (−46%)	67.8 (−29%)	34.0 (−40%) (+13%)	72.5 (−24%) (+7%)	+29% [+36%]
					182 23	18.9	168 19	13.2	147 20.5	19.2	102%

Values are presented in mean or numbers and proportions. VO_2_max, maximal aerobic power; Q, cardiac output; SpO_2_max, arterial oxygenation assessed by finger oximetry at peak exercise; HRmax, heart rate at peak exercise; CaO_2_, arterial oxygen content; Hb, hemoglobin; Hct, hematocrit.

## Data Availability

Not applicable.

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
