# Peer review of "Effects of Acute Exposure and Acclimatization to High-Altitude on Oxygen Saturation and Related Cardiorespiratory Fitness in Health and Disease"

_jcm, 2022, doi:10.3390/jcm11226699_

Round 1
Reviewer 1 Report
The authors wrote a review on the topic of the relationship between VO2max, SpO2 and cardiorespiratory function at high altitude. As the authors point out, there are not many studies in this area. Against this background, the authors summarize findings on VO2max and exercise function at high altitude in elderly people and patients with chronic cardiac or chronic respiratory disease, which have not been discussed widely. Their review was very informative for me. I would point out just a few minor issues.
1. The review describes the findings by dividing them into sections for the elderly and patients with chronic cardiac or chronic respiratory disease. However, the variety of information in the review makes it somewhat difficult to understand the content. I thought it would be important to provide a brief summary of the findings at the end of each section to help readers better understanding.
2. There seem to be relatively many reports on the correlation between BNP and elevation, especially for patients with cardiac disease. There are also relatively many reports on the correlation between pulmonary artery systolic pressure and high altitude. These may be closely related to hypoxia due to high altitude. Wouldn't these correlates with the variation in VO2max at high altitude? Please consider adding these statements if necessary.
Author Response
Reviewer 1.
Dear Reviewer:
Thank you very much for your helpful comments. Please, find our responses below. All changes in the revised manuscript are highlighted (yellow).
The authors wrote a review on the topic of the relationship between VO2max, SpO2 and cardiorespiratory function at high altitude. As the authors point out, there are not many studies in this area. Against this background, the authors summarize findings on VO2max and exercise function at high altitude in elderly people and patients with chronic cardiac or chronic respiratory disease, which have not been discussed widely. Their review was very informative for me. I would point out just a few minor issues.
- The review describes the findings by dividing them into sections for the elderly and patients with chronic cardiac or chronic respiratory disease. However, the variety of information in the review makes it somewhat difficult to understand the content. I thought it would be important to provide a brief summary of the findings at the end of each section to help readers better understanding.
Re: Thank you very much for this important suggestion. We added summary statements in the discussion section from 4.1. to 4.6. as follows:
Ad 4.1.: “In summary, pulse oximetry is easy to use, may provide valuable information on individual responses to high-altitude exposure (acute and during acclimatization) but requires experienced personnel to interpret the findings.
Ad 4.2.: “Taken together, baseline VO2max predicts endurance performance at high altitude, and due to the close association between VO2max and SpO2, any improvement of SpO2 during acclimatization may be helpful, especially in individuals with low baseline VO2max (see figure 2).”
Ad 4.3.: “In summary, altitude related changes in V̇O2max (and SpO2max) may become increasingly important in the elderly due to the declining V̇O2max with aging.”
Ad 4.4.: “In summary, SpO2max might more decline in women sojourning to high altitude, but confirmation of this observation is necessary.”
Ad 4.5.: “Taken together, while patients suffering from CAD or T2DM show rather normal adaptations to high altitude, severely obese individuals seem to be more negatively affected. Importantly, an appropriate CRF (depending on the level of physical activity and altitude expected) is of utmost importance for all these patients intending to visit high-altitude regions, and pulse oximetry (SpO2 and heart rate values) will provide valuable information on acute responses and the progress of acclimatization to high altitude.”
Ad 4.6.: “Taken together, decreases of SpO2 and exercise performance are more pronounced in COPD patients exposed to acute and chronic high altitude when compared to healthy individuals and those with cardiovascular and/or metabolic diseases.”
- There seem to be relatively many reports on the correlation between BNP and elevation, especially for patients with cardiac disease. There are also relatively many reports on the correlation between pulmonary artery systolic pressure and high altitude. These may be closely related to hypoxia due to high altitude. Wouldn't these correlates with the variation in VO2max at high altitude? Please consider adding these statements if necessary.
Re: Thank you for this interesting hint. We added in 4.5.: “Finally, there are several other factors associated with SpO2 and performance, and the risks of AMS and/or cardiovascular adverse events at high altitude, e.g., natriuretic peptides (NPs: BNP/NT-proBNP) are related to both high pulmonary artery pressure and AMS as well [115].”
Thank you very much again.
Reviewer 2 Report
The accumulation effect of high altitude on SpO2 and related VO2max in healthy and diseased individuals was less discussed because the acute effect of hypoxia on this relationship was almost focused. Thus, this manuscript focused on this topic, and I believe that the discussion in this manuscript should provide important information in this research area.
However, I have some questions that may need to be addressed to improve this manuscript.
The rationale and investigating impact of this review paper is unclear. The authors should provide it carefully. The authors focused on the relationship between SpO2 and VO2 (this is a topic in this manuscript) max but its potential benefit for discussing the acclamation effect of high altitude is unclear. It is well known that low SpO2 attenuates VO2max. Thus, discussing the accumulation of VO2 max (SpO2) may be enough for providing recommendations on appropriate preparation/acclimatization strategies for athletes or patients with cardiovascular disease. The authors should clarify the reason why they need to focus on the relationship between SpO2 and VO2 max more clearly because it is the originality of this review paper. Many previous papers have discussed the accumulation effect of high altitude already. For example, SpO2 is easy to measure, thus it may be material to estimate VO2 max, thus the different relationships between acute and chronic effects of hypoxia may be important. At least, the authors should clarify the clear rationale of this review paper in the introduction. The almost discussion may be important in this research area, but unfortunately, it is unclear and it may be difficult for readers to understand the impact of each discussion.
Author Response
Dear Reviewer:
Thank you very much for your helpful comments. Please, find our responses below. All changes in the revised manuscript are highlighted (yellow).
The accumulation effect of high altitude on SpO2 and related VO2max in healthy and diseased individuals was less discussed because the acute effect of hypoxia on this relationship was almost focused. Thus, this manuscript focused on this topic, and I believe that the discussion in this manuscript should provide important information in this research area.
However, I have some questions that may need to be addressed to improve this manuscript.
The rationale and investigating impact of this review paper is unclear. The authors should provide it carefully. The authors focused on the relationship between SpO2 and VO2 (this is a topic in this manuscript) max but its potential benefit for discussing the acclamation effect of high altitude is unclear. It is well known that low SpO2 attenuates VO2max. Thus, discussing the accumulation of VO2 max (SpO2) may be enough for providing recommendations on appropriate preparation/acclimatization strategies for athletes or patients with cardiovascular disease. The authors should clarify the reason why they need to focus on the relationship between SpO2 and VO2 max more clearly because it is the originality of this review paper. Many previous papers have discussed the accumulation effect of high altitude already. For example, SpO2 is easy to measure, thus it may be material to estimate VO2 max, thus the different relationships between acute and chronic effects of hypoxia may be important. At least, the authors should clarify the clear rationale of this review paper in the introduction. The almost discussion may be important in this research area, but unfortunately, it is unclear and it may be difficult for readers to understand the impact of each discussion.
Re: Thank you, we agree. Indeed, the rationale provided was not easy to understand. We now tried to better point out why we aimed to focus on both the relationship between SpO2max and VO2max and the acclimatization effect on SpO2 as well. (1) It is important to understand the impact of SpO2max on VO2max at low and high altitude (in normoxia and hypoxia). (2) SpO2max drops with increasing altitude and VO2max may consequently rapidly become limiting for aerobic exercise performance in hypoxia, particularly in diseased individuals. Thus, acclimatization effects on SpO2 improvements may be crucial to maintain aerobic performance necessary even for everyday activities like walking, climbing stairs, etc. If such activities represent high-intensity activities (close to the individual VO2max), they provoke large cardiovascular responses associated with an increased risk of cardiovascular adverse events; beneficial effects of acclimatization on SpO2 (and related VO2max) could reduce these risks.
We added to the introduction as follows: “The altitude related drop of VO2max in subjects with low baseline CRF, particularly in those with pre-existing acute or chronic diseases, can turn even every day activities like walking or stair climbing into high-intensity (close to the individual VO2max) physical activities associated with an increased risk for cardiovascular adverse events [16,19,21]. Thus, even small improvements of SpO2 (and related VO2max) during acclimatization to high altitude may increase VO2max and reduce the risk of cardiovascular adverse events [10,19,21]. As SpO2 can be easily determined at any place, it provides valuable information on changes of the individual VO2max (and potentially associated risks) during the high-altitude sojourn [16,17,19].”
Thank you very much again.
Reviewer 3 Report
In this narrative review, Furian et al. investigated the effect of high-altitude acclimatization on oxygen saturation and related cardiorespiratory fitness. I have the following comments:
- Did the systematic search include the 3 studies presented in this review?
- How was the further literature on patients found?
- Was the protocol of the systematic review previously registered (e.g. PROSPERO)?
- Table 1 is somewhat confusing. What does the black line represent?
- Compared to the results paragraph on healthy, well-trained individuals, results on patients are more already discussed than the blank results presented.
- Check for abbreviation introducing.
- Are there studies on acclimatization in obese and DM patients?
- The presented studies focus more on acute effects than on acclimatization, especially in obese and respiratory patients. Therefore, the title does not fully cover the included studies.
- Although it is interesting, first part of the discussion does more summarize the overall effects of high.altitude than the acclimatization effects especially in patients.
- Line 381: Please correct sentence: “even during submaximal during exercise”.
Author Response
Dear Reviewer:
Thank you very much for your helpful comments. Please, find our responses below. All changes in the revised manuscript are highlighted (yellow).
In this narrative review, Furian et al. investigated the effect of high-altitude acclimatization on oxygen saturation and related cardiorespiratory fitness. I have the following comments:
- Did the systematic search include the 3 studies presented in this review?
- How was the further literature on patients found?
- Was the protocol of the systematic review previously registered (e.g. PROSPERO)?
Re: Indeed, these questions are justified. Unfortunately, the review was not registered at PROSPERO and efforts made to search more systematically failed, thus we do no longer refer to “systematic search” but simply to the narrative nature of this review. The new method part now reads as follows: “….and was not registered at PROSPERO. PubMed, Web of Science and Google Scholar search engines were used to identify appropriate peer-reviewed studies. Search terms such as “oxygen saturation”, “VO2max”, “exercise capacity”, “hypoxia”, “altitude”, “acclimatization”, “cardiovascular disease”, “coronary artery disease”, “chronic obstructive pulmonary disease”, “obesity”, “type 2 diabetes mellitus” were used to filter studies related to the topic. According to the objectives, pivotal studies have been hand-picked by the authors.
We here present the results of 3 selected studies, likely reporting representative physiological responses on main determinants of VO2max (i.e., maximal values of cardiac output, Qmax; SpO2max and related arterial oxygen content, CaO2max) in healthy subjects. Further studies have been selected to provide more specific information… “
- Table 1 is somewhat confusing. What does the black line represent?
Re: We agree; this is partly due to the change from horizontal to vertical format in the present pdf. The black lines separate the main parameters (now presented in red) and help to differentiate between sea level, acute and chronic hypoxia. Of course, in the case of acceptance we will try to optimize the table in cooperation with the typesetters.
- Compared to the results paragraph on healthy, well-trained individuals, results on patients are more already discussed than the blank results presented.
Re: Thank you for this point. We now have added more data in the result section and deleted “discussion parts”. For example, we added: “In the study by Levine and colleagues, VO2peak values decreased by 12% (24.0±1.2 to 21.1±1.2 mL/min/kg) when acutely exposed to 2,500 m but returned to baseline (24.2±4.8 mL/min/kg) after 5 days of acclimatization [19]. Sea level VO2peak and the ventilatory response to hypoxia (HVR) explained 92% of the variance in VO2peak measured at altitude [19].
Schmid et al. reported a similar altitude-related VO2peak decline (28.3±4.4 to 22.9±3.9 mL/min/kg) in CAD patients exposed to 3,454 m [27]. Exercise responses at altitude, i.e., ventilation, heart rate, and lactate were all significantly higher at comparable levels of exercise [27], indicating elevated exercise stress and cardiovascular risks in CAD patients ascending acutely to high altitude [28,29].”
And: “In that study, maximal power (watts) had decreased from sea level to 4,200 m by about 30% in CAD patients and controls as well [30]”.
- Check for abbreviation introducing.
Re: Yes, done.
- Are there studies on acclimatization in obese and DM patients?
Re: Actually, such studies are scarce. We added as follows: “Ri-Li et al. showed more rapidly increasing AMS scores during the simulated high-altitude (3,658 m) sojourn in obese compared to non-obese men, and this observation was related to lower SpO2 values (median: 75% vs. 83%) in the obese individuals during sleep in hypoxia [30]. Studies on exercise performance and acclimatization in obese people are scarce, but recent data indicate that higher body mass is associated with lower SpO2 values in long-term high-altitude residents [31].”
And: “Individuals suffering from T2DM may have similar adaptations, e.g., with regard to ventilation and cardiovascular responses, but elevated counter-regulatory hormones may negatively affect glycemic control [33].”
- The presented studies focus more on acute effects than on acclimatization, especially in obese and respiratory patients. Therefore, the title does not fully cover the included studies.
Re: Thank you. As suggested (please, see above), we have added some parts on acclimatization effects. We also changed the title to: “Effects of acute exposure and acclimatization to high-altitude on oxygen saturation and related cardiorespiratory fitness in health and disease”.
- Although it is interesting, first part of the discussion does more summarize the overall effects of high.altitude than the acclimatization effects especially in patients.
Re: Yes, agreed. We now included aspects of acclimatization, e.g., as follows: Even small improvements of SpO2 values at rest and at peak exercise, commonly occurring during the first days of acclimatization to high altitude [19,23-26,41], are associated with VO2max improvements [19,23-26,41] and thus, with reduced risk for major cardiovascular events [42-44].
And:
“Beneficial effects of acclimatization occur to a similar extent in healthy individuals and those suffering from cardiovascular diseases [19,26,28]. Those effects remain unclear in obese people and those suffering from COPD, likely related to large differences in individual responses to hypoxia and exercise [31,35].”
- Line 381: Please correct sentence: “even during submaximal during exercise”.
Re: Done as suggested.
Thank you again for the helpful suggestions.
Round 2
Reviewer 2 Report
Thank you for your work.
Reviewer 3 Report
All my comments were addressed accordingly.